# Autonomous Robotic Reinforcement Learning with Asynchronous Human Feedback

**Max Balsells**[1]*    **Marcel Torne**[2]*    **Zihan Wang**[1]*    **Samedh Desai**[1]
**Pulkit Agrawal**[2]    **Abhishek Gupta**[1]
[1]University of Washington    [1]Massachusetts Institute of Technology
{balsells,avinwang,samedh,abhgupta}@cs.washington.edu
{marcelto,pulkitag}@mit.edu

**Abstract:** Ideally, we would place a robot in a real-world environment and leave it there improving on its own by gathering more experience autonomously. However, algorithms for autonomous robotic learning have been challenging to realize in the real world. While this has often been attributed to the challenge of sample complexity, even sample-efficient techniques are hampered by two major challenges - the difficulty of providing well "shaped" rewards, and the difficulty of continual reset-free training. In this work, we describe a system for real-world reinforcement learning that enables agents to show continual improvement by training directly in the real world without requiring painstaking effort to hand-design reward functions or reset mechanisms. Our system leverages occasional non-expert human-in-the-loop feedback from remote users to learn informative distance functions to guide exploration while leveraging a simple self-supervised learning algorithm for goal-directed policy learning. We show that in the absence of resets, it is particularly important to account for the current "reachability" of the exploration policy when deciding which regions of the space to explore. Based on this insight, we instantiate a practical learning system - GEAR, which enables robots to simply be placed in real-world environments and left to train autonomously without interruption. The system streams robot experience to a web interface only requiring occasional asynchronous feedback from remote, crowd-sourced, non-expert humans in the form of binary comparative feedback. We evaluate this system on a suite of robotic tasks in simulation and demonstrate its effectiveness at learning behaviors both in simulation and the real world. Project website https://guided-exploration-autonomous-rl.github.io/GEAR/.

**Keywords:** Autonomous Learning, Reward Specification, Reset-Free Learning, Crowdsourced Human Feedback

## 1   Introduction

Robotic reinforcement learning (RL) is a useful tool for continual improvement, particularly in unstructured real-world domains like homes or offices. The promise of *autonomous* RL methods for robotics is tremendous - simply place a robotic learning agent in a new environment, and see a continual improvement in behavior with an increasing amount of collected experience. Ideally, this would happen without significant environment-specific instrumentation, such as resets, or algorithm design choices (e.g. shaping reward functions). However, the practical challenges involved in enabling real-world autonomous RL are non-trivial to tackle. While those challenges have often been chalked down to just sample efficiency [1, 2], we argue that the requirement for constant human *effort* during learning is the main hindrance in autonomous real-world RL. Given the episodic nature

---

*Denotes equal contribution

7th Conference on Robot Learning (CoRL 2023), Atlanta, USA.

of most RL algorithms, human effort is required to provide constant resets to the system [3, 4, 5], and to carefully hand-design reward functions [6, 7, 8] to succeed. The requirement for constant human intervention to finetune the reward signal and provide resets [3], hinders real-world RL.

The field of autonomous RL [3, 4, 5] studies this problem of enabling uninterrupted real-world training of learning systems. A majority of these techniques aimed to infer reward functions from demonstrations or goal specifications [9, 10, 11, 12], while enabling reset-free learning by training an agent to reset itself [13, 9, 5]. However, these techniques can be challenging to scale to tasks with non-trivial exploration [14].

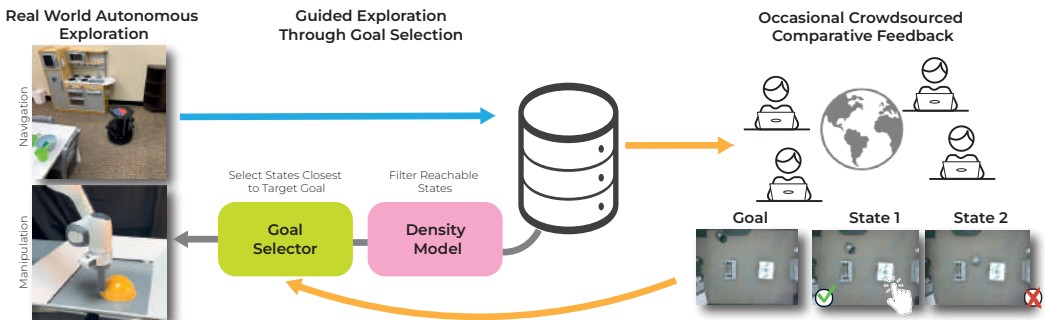

Figure 1: Problem setting in GEAR. The robot explores the world autonomously and reset-free only using cheap, occasional binary feedback from non-expert users to guide exploration. This allows for massive scaling of data experience and solving much more challenging tasks.

To enable autonomous RL techniques to scale to complex tasks with non-trivial exploration, we note that directly reaching a final target goal through autonomous exploration can be difficult. However, achieving a promising intermediate sub-goal can be relatively simple. This process can then be repeated until the desired target goal is accomplished, making the overall exploration process tractable, as long as sub-goals are appropriately chosen. The important question becomes - *"What are promising sub-goals to reach and how can we learn how to reach them within autonomous RL?"*

In this work, we note that a promising sub-goal is one that satisfies two criteria: (1) it is closer to the desired final goal than the current state of the system, and (2) it is reachable from the current state under the current policy. While criterion (1) is challenging to estimate without actually having the task solved beforehand, in this work, we show that asynchronously provided non-expert human feedback in the form of binary comparisons can be used to cheaply estimate state-goal proximity. This estimate of proximity can be paired with a density model for measuring reachability under the current policy, thereby selecting *promising* intermediate sub-goals that satisfy both criteria (1) and (2). Our proposed learning system - **G**uided **E**xploration for **A**utonomous **R**einforcement learning (GEAR) leverages occasionally provided comparative human feedback [15, 16] for directing exploration towards promising intermediate sub-goals, and self-supervised policy learning techniques [17] for learning goal-directed behavior from this exploration data. This is a step towards an autonomous RL system that can leverage both self-collected experience and cheap human guidance to scale to a variety of complex real-world tasks with minimal effort.

## 2 Related Work

**Autonomous Reinforcement Learning in Real World.** RL has been used to learn skills in the real world through interaction with real-world environments [18, 19, 20, 21, 22, 23]. A common limitation is the requirement for episodic resets, necessitating frequent human interventions. "Autonomous reinforcement learning" aims to obviate these challenges by building learning systems that can learn with minimal interventions [4, 24, 5]. A large class of autonomous RL methods, involve learning a forward policy to accomplish a task and a backward policy to reset the system [24, 25, 4, 26, 27, 5]. A different class of methods [13, 28, 29, 30] views the reset-free RL problem as a multi-task RL problem. These techniques typically require a human-provided reward [5, 24, 4, 13, 28] or rely

on simple techniques like goal-classifiers to provide rewards [11, 26]. These reward mechanisms fail to solve domains with challenging elements of exploration. GEAR is able to learn autonomously with no manual resets, while using cheap, asynchronous human feedback to guide exploration more effectively.

**Learning from Human Feedback:** To alleviate the challenges of reward function specification, we build on the framework of learning from binary human comparisons [15, 31, 32, 16, 33, 34, 35, 36]. These techniques leverage binary comparative feedback from human supervisors to learn reward functions that can then be used directly with RL methods, often model-free [37]. While these techniques have often been effective in learning for language models [38, 39], they have seen relatively few applications in fully real-world autonomous RL at scale. The primary challenge is that these methods are too sample inefficient for real-world use, or require too much human feedback [15, 35]. In this work, we build on a recently introduced technique [40] that combines self-supervised policy learning via hindsight supervised learning [17] with learning from human feedback to allow for robust learning that is resilient to infrequent and incorrect human feedback. While [40] was largely evaluated in simulation or in simple episodic tasks, we leverage insights from [40] to build RL systems that do not require resets or careful environment setup.

Some work in robotics that rely on human feedback, scale it up by means of crowdsourcing [41, 42]. We show that GEAR can also work from crowdsourced feedback by using Amazon Mechanical Turk as a crowdsourcing platform, collecting annotations in the form of binary comparisons. Note that other types of human feedback have also been leveraged in previous work, such as through physical contact [43, 44, 45], eye gaze [46], emergency stop [47]. Our method is agnostic to the kind of feedback as long as we can translate it into a sort of distance function to guide subgoal selection.

While it hasn't been tackled in this paper due to the scarce amount of feedback needed in the tested tasks, research done in learning when to ask for human feedback [48, 49, 50, 51, 52, 53] could be leveraged in GEAR to increase efficiency in the amount of feedback requested. Similarly, previous work on shared autonomy and how to improve the understanding of the human/robot intentions [54, 55, 56] could also be applied to GEAR to allow for better use of human feedback.

**Goal-conditioned reinforcement learning:** In this work, we build on the framework of goal-conditioned policy learning to learn robot behaviors. Goal-conditioned RL [17, 57, 58, 59, 60] studies the sub-class of MDPs where tasks involve reaching particular "goal" states. The key insight behind goal-conditioned RL methods is that different tasks can be characterized directly by states in the state space that the agent is tasked with reaching. Based on this insight, a variety of self-supervised goal-conditioned RL techniques have been proposed [58, 61, 62, 63, 59, 17, 64, 65, 66] that aim to leverage the idea of "hindsight"-relabeling to learn policies. These techniques typically rely on policy generalization to show that learning on *actually* achieved goals can help reach the desired ones. As opposed to these techniques, GEAR uses self-supervised policy learning [17, 61] for acquiring goal-directed behavior while relying on human feedback [67] to direct exploration.

## 3 Preliminaries

**Problem Setting.** In this work, we focus on the autonomous reinforcement learning problem [4]. We model the agent's environment as a Markov decision process (MDP), symbolized by the tuple $(\mathcal{S}, \mathcal{A}, \mathcal{T}, \mathcal{R}, \rho_0, \gamma)$, with standard notation [68]. The reward function is $r \in \mathcal{R}$, where $r : S \times A \rightarrow \mathcal{R}^1$ is unknown to us, as it is challenging to specify. As we discuss in Section 4, an approximate reward $\hat{r}(s)$ must be inferred from human feedback in the process of training. As in several autonomous RL problem settings, of particular interest is the initial state distribution $\rho_0(s)$, which is provided for evaluation, but the system is run reset-free without the ability to reset the system to initial states $s \sim \rho_0(s)$ during training [4]. The aim is to learn a policy $\pi : \mathcal{S} \rightarrow \mathcal{A}$, that maximizes the expected sum of rewards $\mathbb{E}_{\pi, \rho_0} \left[ \sum_{t=0}^{\infty} \gamma^t r(s_t, a_t) \right]$ starting from the initial state distribution $\rho_0(s)$ and executing a learned policy $\pi$.

**Goal-Conditioned Reinforcement Learning.** Since reward functions are challenging to define in the most general case, goal-conditioned policy learning techniques consider a simplified class of MDPs where rewards are restricted to the problem of reaching particular goals. The goal-reaching problem can be characterized as $(\mathcal{S}, \mathcal{A}, \mathcal{T}, \mathcal{R}, \rho_0, \gamma, \mathcal{G}, p(g))$, with a goal space $\mathcal{G}$ and a target goal distribution $p(g)$ in addition to the standard MDP setup. In the episodic goal-conditioned RL setting, each episode involves sampling a goal from the goal distribution $g \sim p(g)$, and attempting to reach it. The policy, $\pi$, and reward, $r$, are conditioned on the selected goal $g \in \mathcal{G}$. Goal-conditioned RL problems leverage a special form of the reward function as $r(s, a, g) = \mathbb{1}(s = g)$, 1 when a goal is reached, and 0 otherwise. As in standard RL, at evaluation time an agent is tasked with maximizing the discounted reward based on the goal, $\mathbb{E}_{g \sim p(g), \pi, \rho_0}[\sum_{t=0}^{\infty} \gamma^t r(s_t, a_t, g)]$. Note that, unlike most work in goal-conditioned RL, we are in the autonomous goal-conditioned RL setting, where we do not have access to resets during training.

While goal-conditioned RL makes the reward $r(s_t, a_t, g)$ particularly easy to specify, in continuous spaces $r(s_t, a_t, g) = \mathbb{1}(s_t = g)$ is zero with high probability. Recent techniques have circumvented this by leveraging the idea of *hindsight relabeling* [62, 58, 17]. The key idea is to note that while when commanding a goal the reward will likely by 0, it can be set to 1 had the states that are actually reached, been commanded as goals, thereby providing self-supervised learning signal [62, 58, 17]. This idea of using self-supervision for policy learning has spanned both techniques for RL [58, 59] and iterated supervised learning [61, 17]. The resulting objective for supervised policy learning can be expressed as $\arg\max_\pi \mathbb{E}_{\tau \sim \mathbb{E}_g[\bar{\pi}(\cdot|g), g \sim p(g)]} \left[\sum_{t=0}^{T} \log \pi(a_t|s_t, \mathcal{G}(\tau))\right]$. Self-supervised policy learning algorithms [17] alternate between sampling trajectories by commanding the desired goals under the current policy $\bar{\pi}(\cdot|g)$ and target goal distribution $g \sim p(g)$, where $\bar{x}$ denotes stop-gradient. Policies can then be learned based on the goals reached in hindsight $g = \mathcal{G}(\tau)$, where $\mathcal{G}$ is any function for hindsight relabeling —for instance, choosing the last state of $\tau$ as the goal.

## 4   GEAR: A System for Autonomous Robotic Reinforcement Learning with Asynchronous Human Feedback

Our proposed system - **G**uided **E**xploration for **A**utonomous **R**einforcement learning (GEAR), is able to leverage a self-supervised policy learning scheme with occasional human feedback to learn behaviors without requiring any resets or hand-specified rewards. While typical autonomous RL methods aim to reset themselves autonomously, this comes at the cost of a challenging reward design problem. The reward function has to be able to account for all eventualities that the system may find itself in, rather than behaviors from a single initial state. In this work, we instantiate a practical algorithm GEAR, that takes the perspective that cheap, asynchronous feedback in the form of binary comparisons provided remotely by humans can guide exploration in autonomous RL.

### 4.1   Reset-Free Learning via Goal-Conditioned Policy Learning

The problem of autonomous RL involves learning a policy to perform a task evaluated from a designated initial state distribution $\rho_0$, but without access to episodic resets during training. Typical paradigms for this problem [4, 26, 5] have alternated between training a "forward" policy to solve the task, and a "reverse" policy to reset the environment. The challenge with applying this in the real world boils down to the difficulty of reward specification.

To circumvent the challenge of reward specification, we can model the problem as a goal-conditioned one and leverage self-supervised learning methods [17]. This involves learning a single goal-conditioned policy $\pi(a|s, g)$ that can perform both forward and reverse tasks. While trying to accomplish the forward goal, the policy takes $g \sim p(g)$ as its goal, and once the goal is accomplished, the reverse process takes in $g \sim \rho_0(s)$ as its goal, bringing the agent back to its initial state. The policy $\pi(a|s, g)$ can be learned by self-supervised goal-conditioned policy learning methods [17, 61]. In these approaches, states that were reached in some trajectory during training are rela-

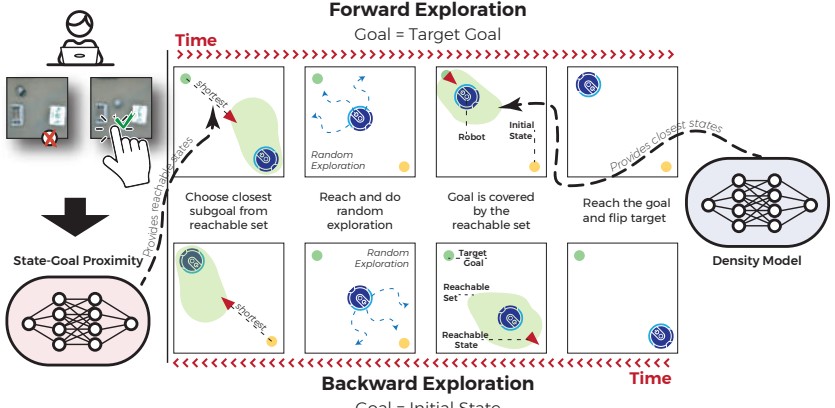

Figure 2: Depiction of autonomous exploration with GEAR - the policy alternates between trying to go to a goal state and getting back to the initial state. In doing so the agent is commanded an intermediate sub-goal that is both proximal to the goal, and reachable under the current policy. When this is absent, the policy performs random exploration. The resulting policy learns to go back and forth, while efficiently exploring the space.

beled, in hindsight, as goal states from which those trajectories serve as examples of how to reach them. Then, they use supervised learning to learn policies from this data (Section 3).

A problem with this kind of self-supervised policy learning algorithm is that it relies purely on policy generalization for exploration; the policy $\pi$ is commanded to reach $g \sim p(g)$ or $g \sim \rho_0(s)$, even if it has not actually seen valid paths to reach $g$. This may result in very poor trajectories since the pair $(s, g)$ can be out-of-distribution. This problem is especially exacerbated in autonomous RL settings.

## 4.2 Guided Exploration and Policy Learning via Asynchronous Human Feedback

Rather than always commanding the initial state or the target goal as described above, human feedback can help select meaningful intermediate sub-goals to command the policy to interesting states from which to perform exploration. In autonomous RL, meaningful intermediate sub-goals $g_{\text{sub}}$ are those that make *progress* towards reaching the currently desired goal. Given a desired goal $g$, the intermediate sub-goal $g_{\text{sub}}$ should be - (1) close to $g$ in terms of dynamical distance [66], and (2) reachable from the current state $s$ under the policy $\pi$. Without knowledge of the optimal value function $V^*$, it is non-trivial to estimate both conditions (1) and (2). In GEAR, we rely on binary feedback provided by human users to estimate state-goal proximity (condition (1)) and on density estimation for computing state-goal reachability (condition (2)).

**Proximity Estimation Using Human Feedback**   To estimate state goal proximity, we draw inspiration from work [40, 15, 16] in learning from human preferences. Specifically, we build directly on a recently proposed technique that uses human preferences to guide exploration in the *episodic* setting [40]. While [40] also guides exploration using human preferences to estimate state goal proximities, it has a strict episodic requirement making it unsuitable for autonomous RL.

Techniques based on comparative feedback aim to learn reward functions from binary comparisons provided by non-expert human supervisors. In this framework, human users can be asked to label which state $s_i$ or $s_j$ is closer to a particular desired goal $g$. These preferences can then be used to infer an (unnormalized) state-goal proximity function $d_\phi(s, g)$ by optimizing $d_\phi$ with respect to the following objective (derived from the Bradley-Terry economic model [67, 15, 33]): $\mathcal{L}_{\text{rank}}(\theta) =$

$$-\mathbb{E}_{(s_i, s_j, g), \sim \mathcal{D}} \left[ \mathbb{1}_{i<j} \left[ \log \frac{\exp -d_\phi(s_i, g)}{\exp -d_\phi(s_i, g) + \exp -d_\phi(s_j, g)} \right] + (1 - \mathbb{1}_{i<j}) \left[ \frac{\exp -d_\phi(s_j, g)}{\exp -d_\phi(s_i, g) + \exp -d_\phi(s_j, g)} \right] \right].$$

This suggests that if a state $s_i$ is preferred over $s_j$ in terms of its proximity to the goal $g$, the unnormalized distance should satisfy $d_\phi(s_i, g) < d_\phi(s_j, g)$.

During exploration, we can choose a sub-goal by sampling a batch of visited states and score them according to $d_\phi(s, g)$. As outlined in [40], the chosen sub-goal doesn't need to be the one that minimizes the estimated distance to the goal, but rather can be sampled softly from a softmax distribution $p(g_{\text{sub}}|s, g) = \frac{\exp{-d_\phi(s,g)}}{\sum_{s' \in \mathcal{D}} \exp{-d_\phi(s',g)}}$ to deal with imperfections in human comparisons.

**Reachability Estimation Using Likelihood Based Density Estimation**  While the aforementioned proximity measure, suggests which intermediate sub-goal $g_{\text{sub}}$ is closest to a particular target goal $g$, this does not provide a measure of "reachability" - whether $g_{\text{sub}}$ is actually reachable by the current policy $\pi$, from the current environment state $s$. This is not a problem in an episodic setting, but in the autonomous reset-free learning case, many states that have been visited in the data buffer may not be reachable from the current environment state $s$ using the current policy $\pi$.

In our self-supervised policy learning scheme (Section 4.1), reachability corresponds directly to marginal density - seeing $(s, g_{\text{sub}})$ pairs in the dataset is likely to indicate that $g_{\text{sub}}$ is reachable from a particular state $s$. This is a simple consequence of the fact that policies are learned via hindsight relabeling with supervised learning [17], and that a supervised learning oracle would ensure reachability for states with enough density in the training set. This suggests that the set of *reachable* intermediate sub-goals $g_{\text{sub}}$ can be computed by estimating and then thresholding the marginal likelihood of various $(s, g_{\text{sub}})$ in the training data. To do so, a standard maximum likelihood generative modeling technique can be used to learn a density $p_\psi(s_t, g_{\text{sub}})$ [69, 70, 71, 72, 73].

The learned density model $p_\psi(s_t, g_{\text{sub}})$ can be used to select *reachable* goals with a simple procedure - given a batch of sampled candidate sub-goals from the states visited thus far, we first filter reachable candidates by thresholding density $p_\psi(s_t, g_{\text{sub}}) \geq \epsilon$. The set of reachable candidates can then be used for sampling a proximal goal proportional to the state-goal distance $d_\phi$ estimated from human feedback as described above. Note that when there are no viable reachable candidates, the policy can perform random exploration. In our experimental evaluation in simulation, we estimate this density with a neural autoregressive density model [69, 70, 71] or a discretized, tabular density model.

## 4.3  System Overview

The overall system in GEAR learns policies in the real world without needing resets or reward functions and minimal non-expert crowdsourced human feedback. GEAR alternates between trying to explore and learn a policy $\pi_\theta(a|s, g)$ to reach the target goal distribution $g \sim p(g)$ in the forward process, and reach the initial state $g \sim \rho_0(s)$ in the reverse one. In each of these processes, the agent selects intermediate sub-goals $g_{\text{sub}}$ for exploration based on the current state and the desired goal. The sub-goal selection mechanism first samples a set of visited states from the replay buffer $\mathcal{D}_{\text{candidates}} = s_{i=1}^N \sim \mathcal{D}$. Then, it uses the density model $p_\psi$ to estimate the reachability of these states from the current one, filtering out the unreachable states. Amongst these, $\mathcal{D}_{\text{candidates}}^{\text{filtered}}$, intermediate sub-goals are sampled $g_{\text{sub}}$ proportional to their estimated proximity to the desired goal, according to the human-trained model $p(g_{\text{sub}}|s, g) = \frac{\exp{-d_\phi(g_{\text{sub}},g)}}{\sum_{s' \in \mathcal{D}_{\text{candidates}}^{\text{filtered}}} \exp{-d_\phi(s',g)}}$. If no states are reachable, the agent performs random exploration. This exploration process repeats until the target goal $g \sim p(g)$(or the start state $g \sim \rho_0(s)$) is reached, then the goal is flipped and the learning continues. Occasionally, the agent updates its density model $p_\psi(s_t, g_{\text{sub}})$ by likelihood-based training and accepts occasional, asynchronous feedback from crowdsourced human supervisors to update the state-goal proximity estimate $d_\phi(g_{\text{sub}}, g)$. To warm-start training, we can add a set of teleoperated data (potentially suboptimal) to the replay buffer, and pretrain our policy $\pi(a|s, g)$ via supervised learning on hindsight relabeled trajectories as described in Section 3 and [17, 61, 40]. Detailed pseudocode and algorithm equations can be found in Appendix A.

## 5   Experimental Evaluation

To evaluate our proposed learning system GEAR, we aim to answer the following questions: **(1)** Is GEAR able to learn behaviors autonomously in simulation environments without requiring resets or

careful reward specification? **(2)** Is GEAR able to scale to learning robotic behaviors directly in the real world using asynchronous crowdsourced human feedback and autonomous RL?

## 5.1 Evaluation Domains

We evaluate our proposed learning system on four domains in simulation and two in the real-world. The benchmarks are depicted in Fig 3 and consist of the following environments: **Pusher**: a simple manipulation task in which we move an object within a table. A **Navigation** task in which a Turtlebot has to move to a certain goal location. **Kitchen**: another manipulation task in which the robot has to open and close a microwave. **Four Rooms**: another navigation task in which an agent has to move through some rooms with tight doors. Details of these environments can be found in Appendix C.

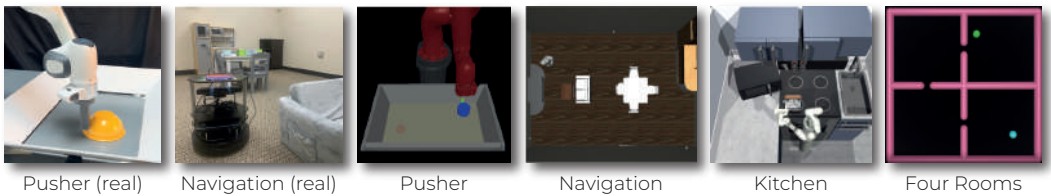

Pusher (real)    Navigation (real)    Pusher    Navigation    Kitchen    Four Rooms

Figure 3: Evaluation Domains for GEAR. We consider a mixture of navigation and manipulation tasks both in simulation and the real world for autonomous learning.

## 5.2 Baselines and Comparisons

To test the effectiveness of GEAR, we compare with several baselines on autonomous reinforcement learning and learning from human feedback. Details of the baselines are presented in Appendix B

## 5.3 Does GEAR learn behaviors autonomously in simulation?

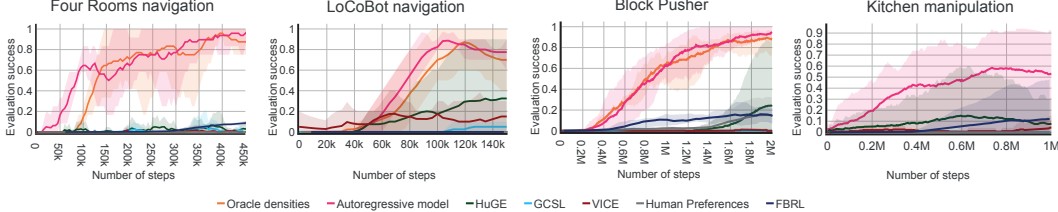

Figure 4: Success rate of autonomous training in simulation of GEAR as compared to baselines. We find that both autoregressive and tabular variants of GEAR are able to successfully accomplish all tasks, more efficiently than alternative reset-free, goal conditioned, and human-in-the-loop techniques

As seen in Fig 4, when evaluation success rates are averaged over 4 seeds, GEAR is able to successfully learn across all environments in simulation. In particular, we see that GEAR outperforms previous work in reset-free RL like VICE or FBRL. In the case of FBRL, this is a consequence of GEAR not relying on a carefully tailored reward function as well as the fact that hindsight relabelling methods are more sample efficient than PPO-based methods [40]. This, together with the more accurate reward signal that GEAR gets from comparative human feedback, explains why GEAR outperforms VICE.

We also see that GEAR is significantly more performant than HUGE, due to the fact that GEAR accounts for policy reachability when commanding subgoals. By ignoring this, the goal selection mechanism in HUGE often selects infeasible goals that get the agent trapped. We explore this topic further in E.1 where we show that GEAR reaches a higher percentage of commanded goals than HuGE. Additionally, by relying on subgoals to guide exploration, GEAR also outperforms GCSL.

Moreover, GEAR beats previous work that rely on human feedback, such as the Human Preferences method, by guiding exploration via subgoal selection and also for the robustness to non-tailored reward signal (feedback) that subgoal selection together with hindsight relabeling brings [40].

Notice that in Fig 4, we conduct experiments using both autoregressive neural density models [69, 70] and tabular densities for measuring policy reachability, except in the kitchen environment, in which the high dimensionality inhibits using tabular densities. While the comparisons in Fig 4 are done from the Lagrangian system state, we will explore scaling this to visual inputs in future work.

In order to obtain a fair comparison between all of the baselines, we leveraged a synthetic oracle instead of a real human. In Section 5.4, we show GEAR succeeds in learning optimal policies when using human feedback in the real world. Furthermore, in Appendix D we show that GEAR can learn successful policies no matter where the feedback is coming from: synthetic oracles, non-expert annotators on Amazon Mechanical Turk, and expert annotators.

In Appendix E, we provide further analysis of the hyperparameters of GEAR, as well as show the effect of the amount and frequency of human feedback needed to learn optimal policies.

### 5.4 Does GEAR learn behaviors autonomously in the real world from crowdsourced human feedback?

Next, we trained GEAR in the real world, learning from crowdsourced human feedback from arbitrary, non-expert users on the web. We set up two different tasks, first, a navigation one with the TurtleBot, and second, a manipulation task with the franka arm consisting of pushing a bowl (Fig 3). The first task involves leaving the robot in a living room and allowing it to explore how to navigate the room autonomously to go from one location to another through obstacles, with human feedback crowdsourced from Amazon Mechanical Turk. The robot is provided with around 10 trajectory demonstrations of teleoperated seed data and then left to improve autonomously. Human supervisors provide occasional feedback: 453 comparative labels provided asynchronously over 8 hours, from 40 different annotators. For the second task, the robot is again provided with 10 trajectories and 200 labels from 22 annotators over the course of one hour. We observe in Fig 5 that our policies successfully learn to solve each task from minimal human supervision and a reset-free setting in the real world . A depiction

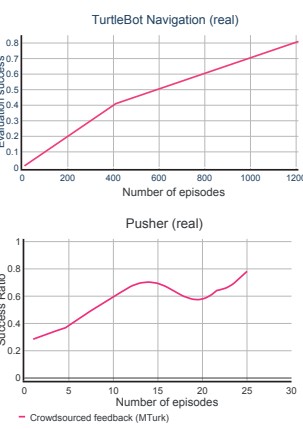

Figure 5: Evaluation in the real-world for Franka pusher and TurtleBot Navigation showing continuous improvement

of the interface can be found in Appendix G, and more details on the demographics of crowdsourced supervision can be found in Appendix D.

## 6   Conclusion and Limitations

In this work, we present a framework for autonomous reinforcement learning in the real world using cheap, asynchronous human feedback. We show how a self-supervised policy learning algorithm can be efficiently guided using human feedback in the form of binary comparisons to select intermediate subgoals that direct exploration and density models to account for reachability. The resulting learning algorithm can be deployed on a variety of tasks in simulation and the real world, learning from self-collected data with occasional human feedback.

**Limitations:** This work has several limitations: (1) *Safety Guarantees:* Real-world exploration with RL can be unsafe, leading to potentially catastrophic scenarios during exploration, (2) *Limitation to Binary Comparisons:* Binary comparisons provided by humans are cheap, but provide impoverished feedback since it only provides a single bit of information per comparison. (3) *Requirement for Pretraining Demonstrations:* For practical learning in the real world, the efficiency of GEAR is enhanced by using teleoperated pretraining data. This can be expensive to collect (4) *Density Model as a Proxy for Reachability:* We use density models as proxies for reachability, but this is only a valid metric in a small set of quasistatic systems. More general notions of reachability can be incorporated. (5) *Learning from low dimensional state:* the current instantiation of GEAR learns from low dimensional state estimates through a visual state-estimation system, which needs considerable tuning.

**Acknowledgments**

We thank all of the participants in our human studies who gave us some of their time to provide labels. We thank the members of the Improbable AI Lab and the WEIRD Lab for their helpful feedback and insightful discussions.

The authors acknowledge the MIT SuperCloud and Lincoln Laboratory Supercomputing Center for providing HPC resources that have contributed to the research results reported within this paper. This research was supported by NSF Robust Intelligence Grant 2212310, the MIT-IBM Watson AI Lab, and the Sony Research Award.

**Author Contributions**

**Max Balsells** led the project and the novel contributions together with Marcel and Zihan. Max and Marcel led the writing. Max led the navigation experiments in simulation and real world and the Kitchen simulated experiments. Max and Marcel led work on the baselines. Max led the work on the ablations.

**Marcel Torne** led the project and novel contributions together with Max and Zihan. Marcel and Max led the writing of the manuscript. Marcel led the pusher simulation and real-world experiments. Marcel led the human crowdsourcing experiments. Max and Marcel led work on the baselines. Marcel was in charge of the figures in the paper.

**Zihan Wang** led the project and novel contributions together with Max and Marcel. Zihan helped in the writing of the paper and conducting ablations.

**Samedh Desai** helped to run the navigation experiments in the real world.

**Pulkit Agrawal** provided some of the valuable resources needed to execute this project.

**Abhishek Gupta** provided guidance throughout the development of the project, assisted with the novel contributions of the paper, and helped in the writing of the paper.

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

Finally, we provide some more insight into our work. In particular, it consists of:

- Appendix A **Algorithm**, shows the pseudocode of our method.

- Appendix B **Baselines**, explains the different baselines that we compared GEAR to in more detail.

- Appendix C **Benchmarks**, explains the different comparison environments in more detail.

- Appendix D **Human Experiments**, gives further details on the human experiments, as data collection procedure and annotators demographics.

- Appendix E **Ablations**, We study the effect of changing key parts of our algorithm or certain hyperparameters.

- Appendix F **Hyperparameters**, depicts the hyperparameters used in the different runs.

- Appendix G **Interface**, show the web interface that we used for collecting human feedback.

## A   Algorithm

We reproduce some of the learning objectives here for posterity. The following is the objective for training the goal selector with human-provided comparative feedback:

$$\mathcal{L}_{\text{rank}}(\theta) = -\mathbb{E}_{(s_i,s_j,g),\sim\mathcal{D}}\left[\mathbb{1}_{i<j}\left[\log\frac{\exp-d_\phi(s_i,g)}{\exp-d_\phi(s_i,g)+\exp-d_\phi(s_j,g)}\right]+ \right. \tag{1}$$

$$\left. (1-\mathbb{1}_{i<j})\left[\frac{\exp-d_\phi(s_j,g)}{\exp-d_\phi(s_i,g)+\exp-d_\phi(s_j,g)}\right]\right] \tag{2}$$

The density model $p_\psi(s_t, g_{\text{sub}})$ can be trained on a dataset $\mathcal{D} = \{(s_t^i, g_{\text{sub}}^i)\}_{i=1}^N$ of relabeled $(s_t, g_{\text{sub}})$ tuples via the following objective:

$$\max_\psi \mathbb{E}_{(s_t,g_{\text{sub}})\sim\mathcal{D}}\left[\log p_\psi(s_t, g_{\text{sub}})\right] \tag{3}$$

Different choices of family for $p_\psi(s_t, g_{\text{sub}})$ yield different variants. We leverage tabular density models and autoregressive. Policies trained via hindsight self-supervision optimize the following objective:

$$\arg\max_\pi \mathbb{E}_{\tau\sim\mathbb{E}_g[\bar{\pi}(\cdot|g),g\sim p(g)]}\left[\sum_{t=0}^T \log\pi(a_t|s_t,\mathcal{G}(\tau))\right] \tag{4}$$

To sample goals from the learned proximity metric, we can sample $g_{\text{sub}} \sim p(g_{\text{sub}}|s,g)$, where

$$p(g_{\text{sub}}|s,g) = \frac{\exp-d_\phi(s,g)}{\sum_{s'\in\mathcal{D}}\exp-d_\phi(s',g)} \tag{5}$$

Below, we show our algorithm in pseudo-code format.

---

**Algorithm 1** GEAR

1: **Input:** Human $\mathcal{H}$, goal $g$, starting position $s$
2: Initialize policy $\pi$, density model $d_\theta$, proximity model $f_\theta$, data buffer $\mathcal{D}$, proximity model buffer $\mathcal{G}$
3: **while** True **do**
4:     $p \sim p(g)$
5:     $\mathcal{D}_\tau \leftarrow \text{PolicyExploration}(\pi, \mathcal{G}, \text{g}, \mathcal{D})$
6:     $\mathcal{D} \leftarrow \mathcal{D} \cup \mathcal{D}_\tau$
7:     $\pi \leftarrow \text{TrainPolicy}(\pi, \mathcal{D})$ (hindsight relabeling [17], Eq 4)
8:     $\mathcal{G} \leftarrow \mathcal{G} \cup \text{CollectFeedback}(\mathcal{D}, \mathcal{H})$ ( Sec 4.2)
9:     $f_\theta \leftarrow \text{TrainGoalSelector}(f_\theta, \mathcal{G})$ (Eq 1 via the Bradley-Terry model [67])
10:    $d_\theta \leftarrow \text{TrainDensityModel}(d_\theta, \mathcal{G})$ (Eq 3, [69, 70, 71])
11: **end while**

---

**Algorithm 2** PolicyExploration

1: **Input:** policy $\pi$, goal selector $f_\theta$, goal $g$, data buffer $\mathcal{D}$
2: $\mathcal{D}_\tau \leftarrow \{\}$
3: $s \leftarrow s_0$
4: **for** $i = 1, 2, \ldots, N$ **do**
5:     **every** k timesteps:
6:     $\mathcal{S} \sim \text{ObtainReachableStates}(d_\theta, s, \mathcal{D})$(Sec 4.2, [69, 70])
7:     $g_b \sim \text{SampleClosestState}(f_\theta, g, \mathcal{S})$(Sec 4.2, Eq 5)
8:     **while** NOT stopped **do**
9:         Take action $a \sim \pi(a|s, g_b)$
10:    **end while**
11:    Execute $\pi_{\text{random}}$ for $H$ timesteps
12:    Add $\tau$ to $\mathcal{D}_\tau$ without redundant states
13: **end for**
14: **return** $\mathcal{D}_\tau$

---

## B  Baselines

We compare GEAR against relevant baselines in autonomous reinforcement learning and learning from human preferences, which are presented next:

- **(1) GCSL [17]:** this baseline involves doing autonomous learning with purely self-supervised policy learning [17], alternating between commanding the start and the goal during exploration

- **(2) HuGE [40]:** this baseline compares with the HUGE algorithm [40], which leverages comparative human feedback asynchronously, but without accounting for policy reachability during training

- **(3) Classifier Based Rewards (VICE) [12, 9]:** this baseline [74, 26] performs forward-backward autonomous RL going between a start position and a goal position with the rewards provided by a classifier that is trained with goal states as positives and on-policy states as negatives, as in [26]

- **(4) Learning from Human Preferences [15]:** this baseline adapts the learning from human preferences paradigm [15] to the autonomous RL setting by commanding goals both forward and backward.

- **(5) Forward Backward RL (FBRL) [24, 75]:** this baseline uses dense reward functions to learn a goal-conditioned policy to reach the goal and go back to the starting set. To evaluate all methods, we follow the protocol in [4], where training proceeds reset-free but intermediate checkpoints are loaded in for evaluation from the initial state distribution.

## C   Evaluation Environments

We briefly discussed the evaluation environments we used to compare our method to previous work. In this section, we will go through the details of each of them.

- **Pointmass navigation:**
  This is a holonomic navigation task in an environment with four rooms, where the objective is to move between the two farthest rooms. This is a modification of a benchmark proposed in [17].

  In this benchmark, the observation space consists of the position of the agent, that is, $(x, y) \in \mathbb{R}^2$, while the action space is discrete of cardinality 9. In particular, there are 8 actions corresponding to moving a fixed amount relative to the current position, the directions are the ones parallel to the axis and their diagonals. Additionally, there is an action that encodes no movement.

  The number of timesteps given to solve this task is 50. Finally, as for a human proxy, we use the distance to the commanded goal, taking into account the walls, i.e., we consider to shortest distance according to the restrictions of the environment.

- **LoCoBot navigation:**
  This benchmark is similar to the Four Rooms one since we are also dealing with 2D navigation. The main difference is that we are working with a simulated robot in Mujoco, in particular a LoCoBot, in a real-life-like environment, in which there is a kitchen and a living room, thus presenting some obstacles for the robot such as tables or a couch. Additionally, the robot works with differential driving, as a LoCoBot or Turtlebot would do. The environment tries to resemble the one we do in the real world with a TurtleBot, so that results obtained in simulation are, to a certain extent, informative about how our robot would perform with the different algorithms in the real world.

  In this environment, the goals the robot should learn how to reach are the lower right and the upper left corners. In this environment, the state space is the absolute position of the robot, together with its angle $(x, y, \theta) \in \mathbb{R}^3$. As we are working with differential driving, the action space is discrete encoding 4 actions: rotate clockwise, rotate counterclockwise, move forward, and no movement.

  The LoCoBot should reach the given goal within 40 timesteps. As before, for the human proxy, we just use the distance to the goal, accounting for obstacles.

- **Block Pusher:**
  This is a robotic manipulation problem, where a Sawyer robotic arm pushes an obstacle to a given location. This benchmark is also a modification of one of the benchmarks proposed by [17]

  In this environment the state space consists of the position of the puck and the position of the arm $(x_1, y_1, x_2, y_2) \in \mathbb{R}^4$. The actions space is the same as in the Pointmass navigation benchmark (i.e. discrete with 9 possible actions).

  The arm should push the object to the desired location in at most 75 timesteps. As for the human proxy, the reward function we use is the following:

  $$r = max(distance\_puck\_finger, 0.05) + distance\_puck\_goal$$

- **Kitchen:**
  This environment is a modification of one of the benchmarks in [4]. It consists of a Franka robot arm with 7 DoF doing manipulation in a kitchen. The objective is to learn how to open and close the microwave.

  The observation space consists of the position of the end-effector of the robot, together with the angle of the microwave joint, that is $(x, y, z, \theta) \in \mathbb{R}^4$. The action space is discrete with cardinality 7, representing moving the end-effector forwards or backwards into any of the three axes, as well as an action encoding no movement. Note that, despite the fact that the

observation space $\subseteq \mathbb{R}^4$, as in the Pusher, the actual range that the values can take is larger than in other benchmarks, and in order to be able to manipulate correctly the microwave, more precision is needed. This is why we couldn't run GEAR with the oracle densities.

The number of timesteps that the Franka has to either open or close the microwave is 100. Finally, when using a human proxy we use the following reward signal:

$$r = \begin{cases} \text{distance(arm, goal arm position)} + |\text{goal joint - current joint}| \, , \, \textit{if success} \\ \text{distance(arm, microwave handle)} + \text{bonus} \, , \, \textit{otherwise} \end{cases} \tag{6}$$

Where by joint we mean the angle of the joint of the microwave, success means that the distance between the current state and the goal state is already below a certain threshold, and the bonus can be any fixed number greater than said threshold.

- **TurtleBot navigation in the Real World:**
  This benchmark is similar to the LoCoBot navigation one, the major difference between the two is that this one takes place in the real world instead of a simulation.

  The goal is to learn how to navigate between two opposite corners in a home-looking environment, with a lot of obstacles. The action and the observation space are the same as in the LoCoBot navigation environment. That is, the action space is discrete with 4 possible actions (move clockwise, counterclockwise, forward, and don't move), while the state space consists of the absolute position of the TurtleBot and its angle $(x, y, \theta) \in \mathbb{R}^3$. In order to get this state, we have a top-down camera and the TurtleBot has blue and red semispheres, whose position can be detected by the camera, thus obtaining the position of the TurtleBot, and its angle (by computing the direction of the vector between the blue and red semispheres of the LoCoBot). Finally, we do collision avoidance by leveraging the depth sensor of the top-down camera.

  The TurtleBot should reach any goal in 25 timesteps. For the human proxy, we just use the Euclidean distance to the goal.

- **Real World Pusher with Franka Panda:** This benchmark is relatively similar to the pusher environment in simulation, except it is with a Franka Emika panda robot in the real world. The goal is to learn how to push an object on the plane between two different corners of an arena. The challenge here is that the pusher is a cylindrical object and planar pushing in this case needs careful feedback control, otherwise, it is quite challenging. The action space is 9 dimensional denoting motion in each direction, diagonals, and a no-op. The state space consists of the position of the robot end effector and the object of interest. In order to get this state we use a calibrated camera and an OpenCV color filter, although this could be replaced with a more sophisticated state-estimation system. The system is provided with *very* occasional intervention when the object is stuck in corners, roughly one nudge every 30 minutes. Success during evaluation is measured by resetting the object to one corner of the arena and commanding it as a goal to reach the other corner.

## D   Human Experiments

We ran GEAR from real human feedback on the Four Rooms navigation environment. We compare the performance of GEAR varying the source of human feedback, coming from a crowdsourced pool of non-expert and expert annotators, and a single expert and non-expert annotator. These experiments were done through the IRB approval of the Massachusetts Institute of Technology.

Qualifications for annotators: We requested certain qualifications from the annotators, some default defined by AMT: Masters qualification. This qualification indicates that the annotator is reliable since has above a threshold of accepted responses. We also defined our own requirements for accepting the responses:

- We required a certain performance on the control task, they had to do better or equal to providing 1 incorrect label and 1 "I don't know" label.

- We required the response to be complete, the annotator should have responded to all questions.

Payment: We paid 0.50$ for a set of 12 questions which take around 2 minutes to answer, which would be equivalent to an hourly pay of 15$/hour.

## D.1 Human Experiments in Simulation

We ran GEAR from real human feedback on the Four Rooms navigation environment. These experiments were run in the span of 4 hours.

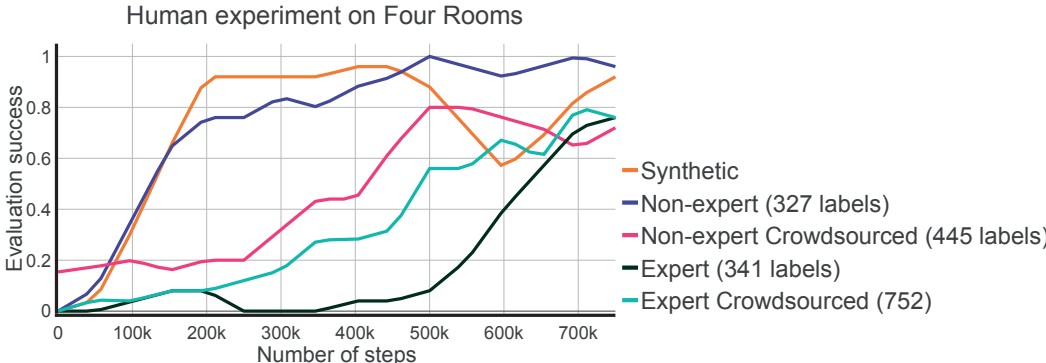

Figure 6: Comparison of GEAR trained from different types of real human feedback. We observe GEAR can be trained with non-expert human feedback without degrading performance.

### D.1.1 Non-expert Crowdsourced feedback

The experiment for this data was collected from annotators on Amazon Mechanical Turk (AMT). There was a total of 78 users who participated. Out of which 12 were discarded because of not meeting the requirements (see Qualifications section above), meaning we collected 445 labels from a total of 66 annotators. The users were presented with the interface shown in Appendix G. They each got a set of 12 questions, 4 on a control task and 8 on the target task. The 4 answers on the control task, where we know the ground truth labels, were used to make sure the annotator understood what the question was and to discard those annotators who did not understand it. As for the demographics, the annotators could optionally respond to a demographics survey. The results are presented in Figure 7.

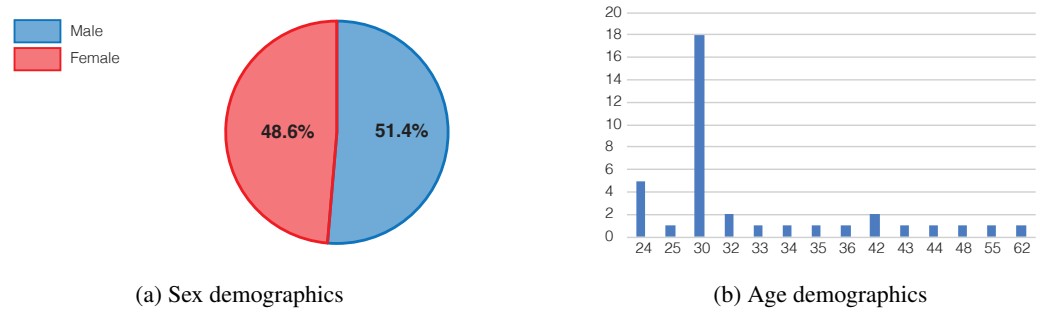

(a) Sex demographics          (b) Age demographics

Figure 7: left: Crowdsource experiment for non-expert annotators sex demographics. right: Crowdsourced experiment for non-expert annotators age demographics

### D.1.2 Expert Crowdsourced Feedback

The experiment for this data was collected with the same interface presented in Appendix G. We recruited the expert annotators through a mailing list in our institution. The annotators were not related to the project, however, they are mostly experts in the fields of computer science and robotics. We collected 752 labels from 29 annotators. We asked the annotators to respond to a demographics survey and the results are presented in Figure 8.

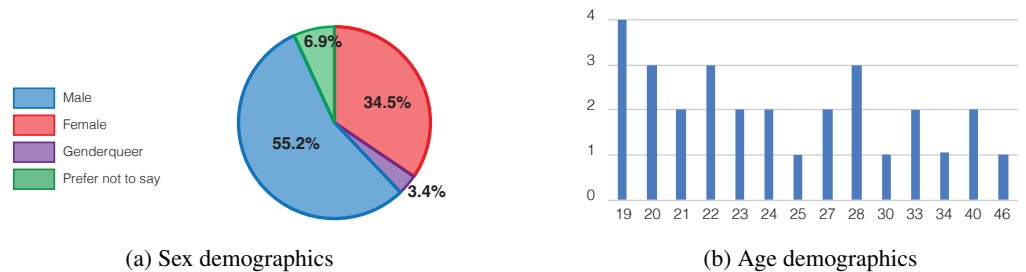

(a) Sex demographics          (b) Age demographics

Figure 8: left: Crowdsourced experiment for expert annotators' sex demographics on the simulated navigation environment. right: Crowdsourced experiment for expert annotators' age demographics on the simulated navigation environment.

### D.1.3 Non-expert feedback

We collected the data from a single annotator, who is an acquaintance of the authors. They are not knowledgeable about either the project or the fields of robotics and computer science. We collected 327 labels using the same interface in Appendix G.

### D.1.4 Expert feedback

We collected data from a single annotator who is knowledgeable in the field of robotics and computer science and is very familiar with the project and the underlying algorithm. We collected 341 labels using the interface in Appendix G.

### D.2 Human Experiments in the real-world

Below we present the demographic statistics of the annotators that provided feedback on the real-world experiments. We note that we left it optional for the annotators to fill in the demographics form, which is the reason why there are fewer data points for the demographics than actual annotators who helped in the experiment.

### D.2.1 Real-world navigation

For the experiment of the pusher in the real world, we collected 453 labels from 40 annotators on Amazon Mechanical Turk.

### D.2.2 Real-world pusher

For the experiment of the pusher in the real world, we collected 200 labels from 22 annotators on Amazon Mechanical Turk.

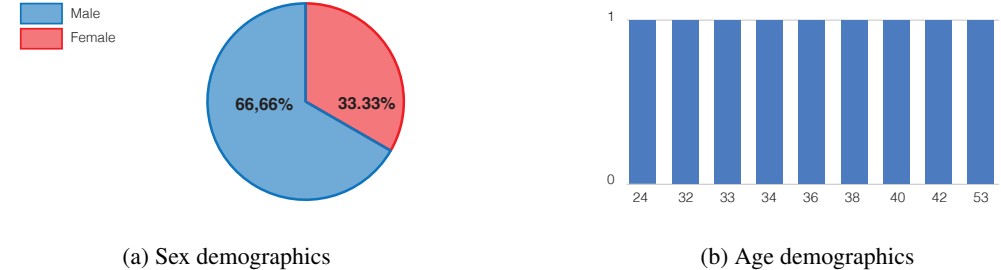

(a) Sex demographics
(b) Age demographics

Figure 9: left: Crowdsource experiment on the real LoCoBot Navigation task for non-expert annotator's sex demographics. right: Crowdsourced experiment on the real LoCoBot Navigation task for non-expert annotator's age demographics

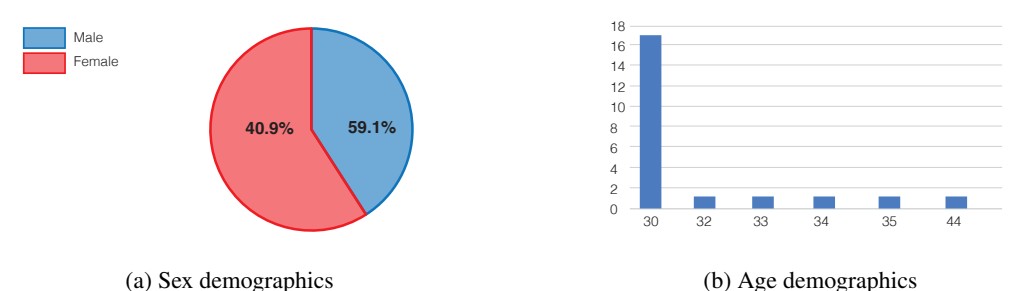

(a) Sex demographics
(b) Age demographics

Figure 10: left: Crowdsource experiment on the real pusher for non-expert annotators sex demographics. right: Crowdsourced experiment on the real pusher for non-expert annotators age demographics

# E   Ablations

## E.1   Analysis of GEAR vs HuGE

In this section, we explore how accounting for reachability makes a crucial difference to the quality of the commanded subgoals. In particular, in Fig 11 we see the percentage of commanded subgoals that are reached in HuGE and in GEAR. Notice that GEAR commands goals every 5 timesteps, while HuGE does so once per episode only, meaning that in HuGE the agent has more time to reach the goal. Despite this, we can clearly see how subgoals in GEAR are clearly reached more consistently than in HuGE, thus, showing how, by accounting for reachability, we manage to command states that are more likely to be reached by the agent.

## E.2   Ablations on the amount of feedback needed

In this section, we study how the frequency at which we provide feedback affects the total amount of feedback or steps we will need to learn a successful policy. In Figure 12 we see that by decreasing the frequency at which we give feedback, we can get a successful policy using fewer queries, that is, less feedback overall. However, when working with low frequencies, we see that the algorithm takes longer to start succeeding. On the other hand, by increasing the frequency at which we provide feedback, we see that the algorithm starts succeeding in terms of timesteps, but we end up needing more annotations overall. So we see a trade-off between the time it takes to succeed and the amount of feedback that we will require.

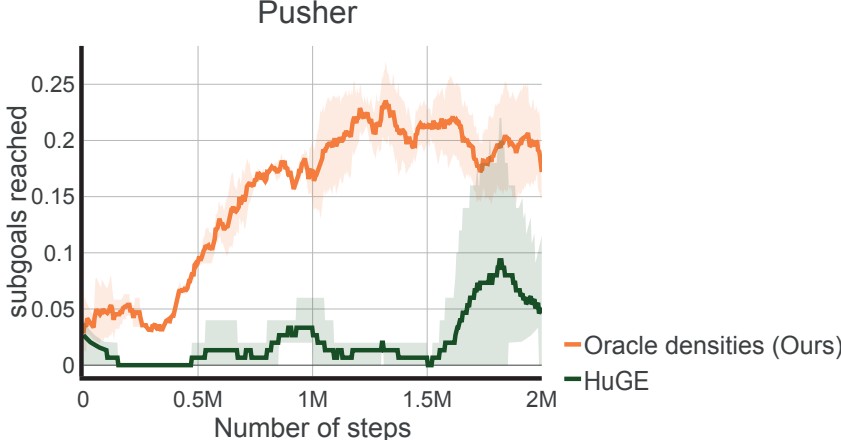

Figure 11: Number of commanded subgoals reached by GEAR (Ours) and HuGE throughout training in the pusher environment. We observe Ours reaches a much higher number of commanded subgoals throughout training, which means that the goals commanded in HuGE are unreachable, because of the reset-free setting, and clearly shows the effect and necessity of introducing reachable sets.

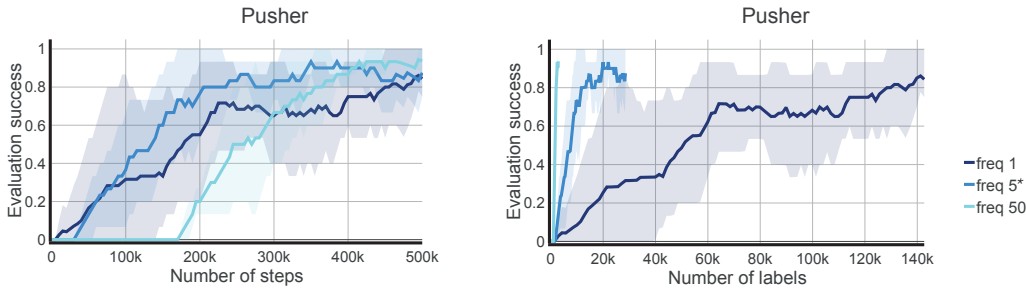

*    *The frequency represents the number of episodes we wait before giving feedback again*

Figure 12: **left:** Comparison of the timesteps needed to succeed depending on the frequency in which we provide feedback. The frequency corresponds to the number of episodes, we wait before giving feedback again. We see that by lowering the amount of feedback, GEAR takes longer to start reaching the goal, however, it still succeeds. **right:** Comparison of the labels needed to succeed depending on the frequency in which we provide feedback. In general, by lowering the frequency of feedback, we can manage to solve tasks with increasingly fewer labels, but as seen in Figure 4, if the frequency is lowered too much, it will have a negative impact on the required number of timesteps to succeed. Hence, we see a tradeoff between the number of labels given and the number of timesteps to achieve the goal

## E.3 Analysis of hyperparameters

To better understand the details of which design decisions affect the performance of GEAR, we conduct ablation studies to understand the impact of various design decisions on learning progress. Specifically, we aim to understand the impact of (1) the threshold $\epsilon$ for the likelihood at which a state is considered "reachable", (2) the frequency at which new intermediate subgoals are sampled during exploration, (3) the algorithm removes redundant exploration steps during exploration, we ablate how important this step is in performance, and (4) we ablate how much pre-training data is required for learning and how this affects learning progress. We find that 1) choosing the right threshold for the success of our algorithm is critical. The best reachable threshold used for the point-mass navigation task is 5. Using a larger threshold (10 or 20) or a smaller one (1) would not make the algorithm work better. 2) the ablation for sampling frequency shows that the right sampling frequency would help boost the performance. We tried sampling frequency with 1, 5, 10, and 20.

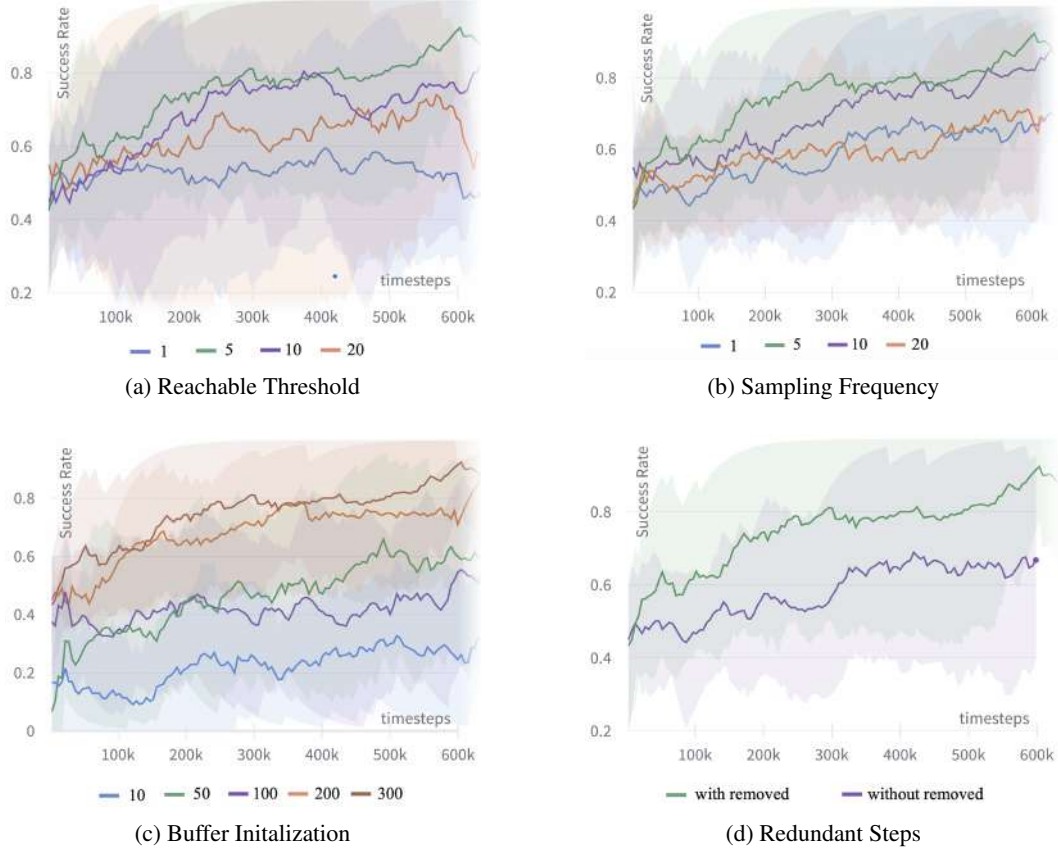

(a) Reachable Threshold

(b) Sampling Frequency

(c) Buffer Initalization

(d) Redundant Steps

Figure 13: Ablations of GEAR: We studied the ablations of GEAR with four different setups. In a), we modify the reachable threshold with a parameter of 1, 5, 10, 20. In b), we verify the effects of sampling frequency. In c), we use a different number of offline data to initialize the buffer. The offline data are collected by driving the agent randomly explore the environment. In d), we evaluate the performance of our algorithm by removing and not removing the redundant exploration steps at each end of the trajectories.

The results show that 5 and 10 have a similar performance. Too small (sampling frequency = 1) or too large (sampling frequency = 20) do not work well. If the sampling frequency is too small, the agent might not be able to reach the subgoal and too frequent subgoal selection would make the performance drop. If the sampling frequency is too large, there would be more redundant wandering steps which make the learning less efficient. 3) We found that removing the redundant steps would help training significantly. Without removing the redundant steps in the trajectory sampling, there would be stationary states when the agent is stuck in the environment which could lead to the drop of performance. 4) More random pre-trained data would help build up the reachable set and further improve the performance.

## F Hyperparameters

In this section, we state the primary hyperparameters used across the different experiments. All the values are shown in Table 1

| Parameter | Value |
|---|---|
| *default (to those that apply)* | |
|     Optimizer | Adam [76] |
|     Learning rate | $5 \cdot 10^{-4}$ |
|     Discount factor ($\gamma$) | 0.99 |
|     Reward model architecture | $MLP(400, 600, 600, 300)$ |
|     Use Fourier Features in reward model | True |
|     Use Fourier Features in policy | True |
|     Use Fourier Features in density model | True |
|     Batch size for policy | 100 |
|     Batch size for reward model | 100 |
|     Epochs policy | 100 |
|     Epochs goal selector | 400 |
|     Train policy freq | 10 |
|     Train goal selector freq | 10 |
|     goal selector num samples | 1000 |
|     Stop threshold | 0.05 |
| *LoCoBot navigation* | |
|     Stop threshold | 0.25 |
| *TurtleBot navigation in Real World* | |
|     Stop threshold | 0.1 |
|     policy updates per step | 50 |
| *Oracle Densities* | |
|     reachable threshold | 5 |
| *VICE* | |
|     reward model epochs | 20 |
| *Human Preferences* | |
|     reward model epochs | 20 |
| *Autoregressive* | |
|     reachable threshold | 0.25 |
|     Epochs density model | 30000 |
|     Train autoregressive model freq | 300 |
|     Batch size for the density model | 4096 |

Table 1: Hyperparameters used when GEAR

## G    Web Interface for Providing Feedback

In Figure 14 we show an example interface for providing feedback for the TurtleBot navigation task, the same is used for the pusher task.

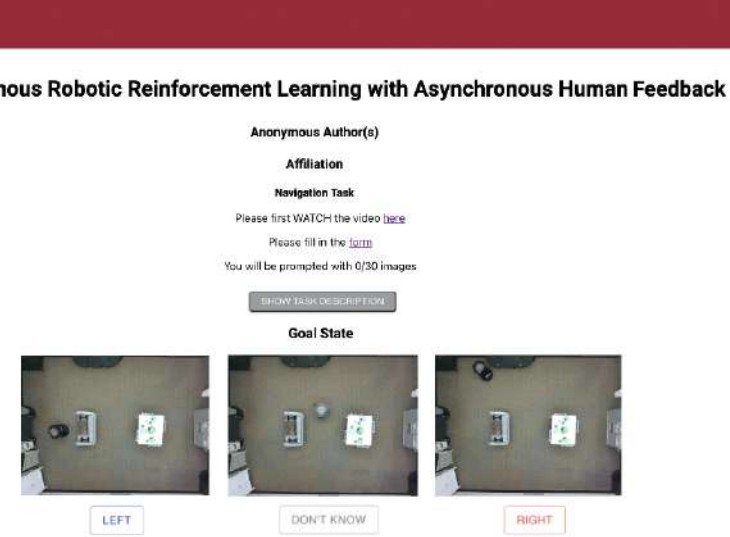

Figure 14: Visualization of the human supervision web interface to provide feedback asynchronously during robot execution. Users are able to label which of the two states is closer to a goal or say they are unable to judge.

