# OpenReview forum: "Autonomous Robotic Reinforcement Learning with Asynchronous Human Feedback"
_robot-learning.org/CoRL/2023/Conference — CoRL 2023 Poster_

### Official Review · Reviewer_yNZB · 2023-07-13

**Confidence:** 4
**Originality:** Good
**Technical Quality:** Very Good
**Clarity Of Presentation:** Very Good
**Impact:** 3

**Recommendation:**

Weak Accept: I recommend accepting the paper, but will not argue for my recommendation if the majority of other reviewers have a different opinion.

**Review:**

The paper is well written and clear.

However, I the claims are stronger than the contributions. I do not agree that the approach solves the world-reset problem, since it is applicable only if the actions that the robot exert in the world are reversible. In simple navigation cases the robot can (often) go back, or the robot can push the object back to a previous position, but in cases that the world change irreversibly, e.g. pouring liquid in a glass, the proposed approach is not applicable. As such, I would not claim that the world-reset problem is addressed, but rather that the approach is only applicable in cases where the actions are reversible.

The presentation could be improved if algorithms and diagrams were provided, giving a better idea of how things work and what are the precise contributions of the paper.

Figures 4 and 5 seem to be trimmed before the convergence and success rate convergence, which makes the results incomplete. Specially Fig. 4, where we can see the figures are trimmed in different points (without any explicit reason) on the x axis. Also there are many more tests, and the results in Figure 5 are presented just for a couple of tests.

Another good point against the paper is the lack of source code, datasets, and the empty website. If there are simulations we should be able to run some tests and evaluate hyper-parameters effects, which would greatly improve the understanding and the strength of the paper.

The Achilles heel of the proposed approach is that binary feedback is know for being best for refining models, applying this in the RL directly is not a good idea. As such, I would recommend to have a learned model in the tests, and see how much faster the model is refined with the binary feedback. Maybe that could be the "niche" of the approach. No modifications would need to be done to the paper for that as well, it is just a matter of testing.

**Quality Of The Limitations Section:**

Limitations are addressed clearly

**Questions For Rebuttal:**

I would like to note that 1900 positive/negative feedback is not really practical, however, reinforcement learning is know for those large numbers. Could you share how this amount of feedback compare to other methods in RL?

**Robotics Focus:**

Relevant but unlikely to deploy to hardware in near future

**Summary Of Paper:**

This paper presents a Reinforcement Leaning (RL) method, which uses asynchronous human feedback in a binary form (positive or negative), which is a very interesting, and novel way which brings knowledge from the learning from preferences framework to RL. An special attention is given to the world-reset problem in RL of real robots, which here is tackled by learning a monolithic forward and inverse model, instead of one model for the former and another for the later. The paper is tested in simple virtual and real tasks of going towards a goal and pushing an object towards a goal.

**Summary Of Recommendation:**

The approach is unlikely to be deployed in hardware in the future due to the still very large amount of trials and human feedback necessary to make a practical working solution.

The main limitations are the still high number of training steps and feedback required, and the limitation to scenarios which the world is reversible. But those were not mentioned.

---

### Official Review · Reviewer_SMKn · 2023-07-17

**Confidence:** 4
**Originality:** Good
**Technical Quality:** Good
**Clarity Of Presentation:** Good
**Impact:** 3

**Recommendation:**

Weak Accept: I recommend accepting the paper, but will not argue for my recommendation if the majority of other reviewers have a different opinion.

**Review:**

### Strengths

-   Overall, the paper is well-structured, and it clearly presents the problems the authors want to address using each specific component of the proposed learning system.
-   The authors present experiments on 3 different environments, dealing with holonomic and differential driving navigation and a manipulation problem. In the last two cases, they also present real-world experiments, using TurtleBot and Franka Panda manipulator.
-   The authors present an ablation study in which they report performances obtained by changing several hyperparameters and by not considering the part of the algorithm in which redundant exploration steps are removed during exploration.
-   The experimental results show an improvement in performance on all the tested environments with respect to the considered baselines in the simulated environments. The approach is able to

### Weaknesses

1.  As a minor suggestion, the language could be slightly improved in the introduction. Avoiding present perfect tenses, such as has looked and has aimed, when possible, and the modal verb “can” followed by the passive form would make the reading flow more smoothly.
2.  There are some inconsistencies throughout the text. Concerning the proposed approach, the authors sometimes say that it is “reset-free” or that it “does not require resets”, while other times it is said that it requires “minimal manual resets”. This point should be clarified.
3.  The authors should better explain the differences between their method and [37], which is a work under review, not available as a preprint. This should be better clarified to evaluate the novelty of this work. Indeed, in section 2, the authors say “In this work, we build on a recently introduced technique [37] that combines self-supervised policy learning via hindsight supervised learning [17] with learning from human feedback to allow for robust learning that is resilient to infrequent and incorrect human feedback. While [37] was largely evaluated in simulation or in simple episodic tasks, we leverage insights from [37] to build RL systems that do not require resets or careful environment setup.” The fact that the proposed method is reset-free and is tested in real-world scenarios seems to be the main difference. However, in section 5.2, [37] is used as a reference for HUGE **reset-free** baseline. In this same section, the authors state that HUGE “leverages comparative human feedback asynchronously, but without accounting for policy reachability during training”. So, is the latter the only difference between your proposed approach and [37]? If the answer is yes, the authors should pose more attention to this last aspect while presenting their approach.
4.  The authors should include a more in-depth discussion of the obtained results. Why does the innovative feature of GEAR with respect to HUGE (density estimation?, see point 3) bring a much more significant performance improvement in LoCoBot and PointMass navigation than in the Block Pusher environment? Why does the autoregressive neural density model work badly in the Block Pusher environment, while it generally reaches higher performances than the discretized densities in the other two environments? Which of these two models did you use to obtain the results shown in Table 5 for real-world environments?
5.  The authors should better clarify how the amount of human feedback is determined and how it influences performances. In the TurtleBot environment, 1900 labels are provided over 6 hours, which is approximately 317 labels per hour. This can still be considered a relatively high amount of human feedback. Could it be reduced without significantly impacting performance? Otherwise, this should probably be added in the limitations section.
6.  The authors use actual human feedback, and not a synthetic oracle, only in the TurtleBot real-world environment. A comparison between the results obtained using the synthetic oracle in this scenario would have been interesting.

**Quality Of The Limitations Section:**

Limitations are addressed clearly

**Questions For Rebuttal:**

1.  The authors should better explain the differences between their method and [37], which is a work under review, not available as a preprint.
2.  The authors should include a more in-depth discussion of the obtained results.
3.  The authors should provide a comparison of performances obtained by increasing or reducing the amount of human feedback.

**Robotics Focus:**

Sufficient demonstration on hardware

**Summary Of Paper:**

This paper proposes Guided Exploration for Autonomous Reinforcement learning (GEAR). Existing approaches for autonomous RL learn forward and backward policies to solve the task and reset the environment respectively, posing the challenge of reward specification. To avoid this problem, GEAR uses self-supervised learning to learn a goal-conditioned policy. However, rather than alternatively sampling the target goal or the initial state, GEAR identifies promising sub-goals. Such goals are selected leveraging occasional asynchronous human feedback in the form of binary comparisons to estimate state-goal proximity and density estimation to compute state-goal reachability given the current policy and environment state.

**Summary Of Recommendation:**

The paper requires improvements on the weaknesses highlighted in the review. Also, contributions are not as clear as they should be.

---

### Official Review · Reviewer_epUg · 2023-07-20

**Confidence:** 4
**Originality:** Good
**Technical Quality:** Fair
**Clarity Of Presentation:** Excellent
**Impact:** 3

**Recommendation:**

Weak Accept: I recommend accepting the paper, but will not argue for my recommendation if the majority of other reviewers have a different opinion.

**Review:**

Strengths:
- This paper tackles an important problem for real-world RL (especially real-world human-in-the-loop RL).
- The authors show that their method is effective in several real and simulated tasks with simulated human feedback
- The paper is mostly well-written, although lacking detail in some places, especially regarding the user testing

Weaknesses:
- The related work is very narrow; 1/3 of the cited works (19/56) are from the same group.  I encourage the authors to read and cite much more widely, especially on the human-robot interaction (HRI) side (e.g. Andrea Thomaz, Julie Shah, Sidd Srinavasa, and their students and postdocs).  Work in HRI may also help to provide better support for how well this method would work in the real world, since it includes testing with non-experts, crowd workers, and other realistic user populations and provides insights into how real feedback deviates from "synthetic proxies".
- There is no information on the human evaluation (in appendix section B the implication is that a proxy is used "And for the human proxy we just use euclidean distance to the goal).  Was the study approved by an ethics board (e.g., an IRB)? Who were the participant(s)? Did they include robotics experts or research team members? The results in 5.4 are impossible to interpret without this information -- for example, if the feedback was provided by research team member, it is possible that the team member used their knowledge of the approach to provide more useful feedback than a typical human teacher would provide.
- Did the pusher environment use a proxy? If so, is not appropriate to include it in section 5.4 (or the title of 5.4 needs to be changed), since it implies that both tests in that section were conducted with human feedback.

Minor points:

- Binary feedback often means a "good" or "bad" (or thumbs up/down) label on a single state; the authors may want to refer to their feedback mechanism as "binary preference" or more consistently as "comparative feedback" to be clear

**Quality Of The Limitations Section:**

Additional details required

**Questions For Rebuttal:**

See above; I would like to see the authors clarify the methods used in the human subjects validation and discuss how they will expand their related work to cover more related work in HRI.  They may wish to also explain how they will update the limitations to address the limitations of the validation.

**Robotics Focus:**

Sufficient demonstration on hardware

**Summary Of Paper:**

This paper extends a prior method for learning using human feedback to guide exploration by adding a reachability model and validating it in robot learning without resets.

**Summary Of Recommendation:**

Overall, this work has interesting technical elements, but the claim that this would actually work with users is not well supported by the validation as currently presented.  Because the method is otherwise a valuable contribution to CoRL, I am rating this as a weak accept, but I would like to see the authors explain in the rebuttal how they will be more careful about their claims (especially if the human subjects validation was not conducted using appropriate experimental techniques).

Post rebuttal comment: I would still rate the validation as one of the weaker elements of the paper, and I'm not at all sure how the authors will pull all the material together into a paper.  I keep my score as a weak accept.

---

### Official Review · Reviewer_oFcp · 2023-07-23

**Confidence:** 3
**Originality:** Good
**Technical Quality:** Good
**Clarity Of Presentation:** Good
**Impact:** 2

**Recommendation:**

Weak Accept: I recommend accepting the paper, but will not argue for my recommendation if the majority of other reviewers have a different opinion.

**Review:**

strengths：
1.The approach of using human feedback to select subgoal is an interesting work.
2. The paper is well-written and its main content is relatively easy to understand.
3. The experiments in the paper are comprehensive, involving various simulations and real-world robot experiments.
weaknesses：
1.In my opinion, the paper seems to focus more on goal-conditioned policy learning rather than reset-free robot learning. The methods and experiments primarily address the problem of subgoal selection, without adequately considering the impact of the initial state distribution in a reset-free setting.
2.The experimental setup and tasks appear to be relatively simple. In reset-free robot learning, the main concern is the influence of changes in the environment due to the task execution on its subsequent learning process. The experiments in the paper do not seem to fully capture this problem.
3.The baseline used for comparison lacks the reset-free methods.
4. When using human feedback for subgoal selection, it is necessary to estimate the distance from two states to the goal. However, the human proxy provided in the appendix is defined as a reward function. It is unclear how this reward function is used to evaluate the distance between states and the goal.

**Quality Of The Limitations Section:**

Additional details required

**Questions For Rebuttal:**

Please check the weaknesses in Review.

**Robotics Focus:**

Sufficient demonstration on hardware

**Summary Of Paper:**

This paper proposes an autonomous reinforcement learning method based on guided exploration to address the reset-free robot learning problem, by incorporating goal-conditioned policy learning and human feedback.

**Summary Of Recommendation:**

In this paper, a framework for autonomous reinforcement learning in the real world using cheap, asynchronous human feedback.  Some weaknesses such as the impact of the initial state distribution in a reset-free setting, the experimental baselines and so on should be addressed.

---

### Author Response · Authors · 2023-08-11
**General comment to the reviewers**

Dear reviewers,

First of all, thank you for your feedback! Some very insightful comments have been pointed out, and we believe that by addressing them we will manage to better show the impact that GEAR can have on the community. Secondly, we are still running some experiments to cover all the concerns of your review, but in order to give more time for an open discussion about our work, we will start sharing some of them.

Notation: we refer to the figures in the additional material, attached to each rebuttal personal response, as Figure AM ${figure_number}

We also invite you to see the updated project [website](https://sites.google.com/view/gearcorl?usp=sharing). And we cleaned and provided the **code** attached to each reviewer's.

Some reviewers asked about learning more on the related unpublished work (HuGE baseline, [37]). We provide the reviewers with an anonymized pre-print of this work.

Note: The results of the experiments we ran during the rebuttal period, the code for GEAR and the pre-print of HuGE are all attached to each one of the personal rebuttal responses.

Please find below a list of the experiments that we are presenting now, more will come on the coming days:
1. GEAR with autoregressive density variant on pusher (Fig. AM 2)  [reviewer SMKn]
- We added the curve for GEAR with the autoregressive density on the pusher benchmark
2. Ablation on amount of human feedback (Section AM 3) [reviewer oFcp, SMKn]
- We analyze how the performance is affected depending on the frequency and number of labels provided by a proxy human feedback.
3. Improved comparison of GEAR vs baselines on the pusher (not using demos) in Figure AM2 [reviewer SMKn]
- In the results presented in the paper in Figure 4 for the pusher benchmark, all baselines were pretrained with demonstration, we changed this and trained all from scratch to emphasize the benefits of GEAR over the other baselines.
4. Running Locobot navigation on sim for more timesteps in Figure AM 1  [reviewer yNZB]
- The Figure 4 for the LoCoBot sim result was trimmed and we present the results on longer timesteps to show they converged.

We are in the process of running the following experiments:
1. More experiments on collecting feedback from non-expert humans in simulation [reviewer epUg]
- We will provide a comparison on training policies from expert annotators against non-expert annotators
2. One more reset-free baseline [oFcp]
- We are implementing the (reset-free) forward-backward RL benchmark
3. One more benchmark (from EARL AM[1]) [reviewer oFcp]
- We are implementing the kitchen benchmark from EARL AM[1] and will run GEAR and the baselines

AM[1] EARL: Environments for Autonomous Reinforcement Learning by Archit Sharma et al, 2021. [arXiv:2112.09605](https://arxiv.org/abs/2112.09605)

---

> ### Author Response · Authors · 2023-08-14
> **Finished experiments**
>
> Dear reviewers,
>
> Thank you once again for taking the time to review our work, we are most appreciative for the effort. We have been answering your remarks during the rebuttal period and we have finished addressing them all. In each reviewer’s rebuttal responses you can find an attached pdf (RebutalExperiments.pdf) with all the additional experiments that were conducted under the rebuttal period.
>
> We are aware the end of the rebuttal time is looming, but we sincerely hope and we would really appreciate it if you were able to engage in a conversation about our work before the deadline.

---

### Decision · Program_Chairs · 2023-08-30

**Decision:**

Accept (Poster)

**Comment:**

The paper initially got mixed reviews as the reviewers were not satisfied with some claims in the paper regarding "reset-free" RL as well as missing details for the human experiments. Moreover, the experiments were conceived to be rather simple even though a large range of environments/tasks have been used. The authors did a good job in their rebuttal and could convince all reviewers to change their score such that we have now an overall positive evaluation. I agree with the reviewers that the paper is now in a much better state given the new experiments. It will however be hard to add this new material into a consistent representation of the paper. I nevertheless would rather to see the paper accepted as the approach and the presented results are very interesting.